# Menthol Increases Bendiocarb Efficacy Through Activation of Octopamine Receptors and Protein Kinase A

**DOI:** 10.3390/molecules24203775

**Published:** 2019-10-20

**Authors:** Milena Jankowska, Justyna Wiśniewska, Łukasz Fałtynowicz, Bruno Lapied, Maria Stankiewicz

**Affiliations:** 1Animal Physiology and Neurobiology, Faculty of Biology and Environmental Protection, Nicolaus Copernicus University, Lwowska 1, 87-100 Toruń, Poland; stankiew@umk.pl; 2Plant Physiology and Biotechnology, Faculty of Biology and Environmental Protection, Nicolaus Copernicus University, Lwowska 1, 87-100 Toruń, Poland; jwisniew@umk.pl (J.W.); lfalt@doktorant.umk.pl (Ł.F.); 3Centre for Modern Interdisciplinary Technologies, Nicolaus Copernicus University, Wileńska 4, 87-100 Toruń, Poland; 4Laboratoire SiFCIR UPRES EA 2647/USC INRA 1330, Université d’Angers, UFR Sciences, 49045 Angers, France; bruno.lapied@univ-angers.fr

**Keywords:** menthol, essential oils, bendiocarb, carbamates, octopamine receptor, protein kinase A, PKA, protein kinase C, PKC

## Abstract

Great effort is put into seeking a new and effective strategies to control insect pests. One of them is to combine natural products with chemical insecticides to increase their effectiveness. In the study presented, menthol which is an essential oil component was evaluated on its ability to increase the efficiency of bendiocarb, carbamate insecticide. A multi-approach study was conducted using biochemical method (to measure acetylcholinesterase enzyme activity), electrophysiological technique (microelectrode recordings in DUM neurons in situ), and confocal microscopy (for calcium imaging). In the electrophysiological experiments, menthol caused hyperpolarization, which was blocked by an octopamine receptor antagonist (phentolamine) and an inhibitor of protein kinase A (H-89). It also raised the intracellular calcium level. The effect of bendiocarb was potentiated by menthol and this phenomenon was abolished by phentolamine and H-89 but not by protein kinase C inhibitor (bisindolylmaleimide IX). The results indicate that menthol increases carbamate insecticide efficiency by acting on octopamine receptors and triggering protein kinase A phosphorylation pathway.

## 1. Introduction

The impact of synanthropic insect species on human beings is highly significant and raises with global warming. The demand for food is increasing with a growing human population in the world, while insect pests destroy growing plants and stored crops. Therefore, new methods of protecting man and food against insect pests are constantly being developed [1,2].

Although conventional chemical insecticides are the most effective in controlling insect pest, they have many disadvantages. First, they are increasingly becoming less efficient. Long- lasting and excessive use have caused the development of resistance to different groups of insecticides (e.g., nicotinoids and pyrethroids) in various insect species [3,4,5]. Cross-resistance (i.e., the specific resistance for given insecticide can also cause resistance for other insecticidal substances) is also observed [5]. In consequence, it is necessary to use increasing doses of insecticides, which in turn leads to growing incidence of environmental pollution and danger to non-target organisms including humans [6,7,8].

Different strategies are used in Integrated Pest Management to increase the efficiency of insecticides and to reduce their dosages. One of them is the application of mixture of insecticides with different molecular targets [9,10,11,12]. Quite a new strategy has been proposed: increase the sensitivity of “classical insecticide targets” by synergistic agents which elevate intracellular calcium concentration and activate intracellular signaling pathways [13,14,15,16]. Potentiating effect of DEET repellent to carbamate insecticides by the activation of metabotropic receptors has been demonstrated [17] and a corresponding mechanism has been proposed for an essential oil component—menthol [18].

Essential oils derived from plants are proposed as alternative substances to chemical insecticides which are safer for animals and humans [19,20,21]. Even though they are much less efficient than classical insecticides and therefore cannot replace them [22]. However, it has been shown that some essential oil components activate the metabotropic octopamine receptors [23,24,25]. They can be especially interesting in potentiating the effects of insecticides because their target, octopamine receptors, play a key role in physiology of insects [26,27].

Octopamine receptors belong to a family of metabotropic G protein-coupled receptors. Their classification was made based on similarities between *Drosophila melanogaster* receptors and mammals’ adrenergic receptors. In insects, the activation of receptors OCTβ-R triggers (*via* the increase of cAMP) protein kinase A (PKA) signaling pathway while the activation of OCTαR (*via* the increase of IP3) triggers protein kinase C (PKC) pathway [26,28]. In *Periplaneta americana,* one octopamine receptor has been characterized—Paoa_1_—and its activation leads to increases in both intracellular calcium level and cAMP level [29].

In our previous studies, it was demonstrated that in the essential oils component, menthol, increases the efficiency of carbamate insecticide (bendiocarb), most probably by activating octopamine receptors [18]. Carbamates are inhibitors of acetylcholinesterase (AChE) enzyme, which hydrolyzes acetylcholine neurotransmitters. Acetylcholine activates synaptic and non-synaptic acetylcholine receptors in the central and peripheral nervous systems of many organisms including insects [30,31]. Increased sensitivity of AChE to carbamate insecticides via activation of metabotropic (muscarinic) receptors has already been shown [17]. The aim of our study was to verify the hypothesis that menthol potentiates the effect bendiocarb through the activation of octopamine receptors and thus the activation of phosphorylation cascade. Bendiocarb, the most often used carbamate insecticide, was approved by World Health Organization and European Union Commission [32,33].

Octopaminergic dorsal unpaired median (DUM) neurons of the cockroach (*Periplaneta americana*) nervous system was taken as a model. DUM neurons release octopamine from their axonal endings and express octopaminergic receptors on the cell body [34,35,36]; they have endogenous pacemaker activity that is precisely regulated by a large variety of ion channels and receptors. DUM neurons are often used as very sensitive models to study the intracellular signaling systems [13,17]. We conducted multi-approach analyses, including electrophysiological experiments on DUM neurons in situ, biochemical tests of AChE activity, and calcium imaging using fluorescent microscopy techniques. Presented study confirms our hypothesis that menthol increases the efficiency of bendiocarb by activation of octopamine receptors, activation of protein kinase A (PKA) signaling pathway, and increasing Ca^2+^ concentration.

## 2. Results

### 2.1. Electrophysiological Tests

Electrophysiological experiments were performed on DUM neurons in situ in ganglia, using the microelectrode technique. DUM neurons display spontaneous activity and generate rhythmical action potentials. Typical DUM neuron interspike “resting potential” is about −50 mV and the discharge frequency is about 2 Hz.

In control conditions, application of physiological saline directly on ganglion caused a short disturbance of DUM neurons activity, mostly manifested in small depolarization (no more than 2 mV) and increase in firing frequency (however with duration no longer than 3 s).

Tested substances changed spontaneous activity pattern in DUM neurons (F_5,45_ = 5.65, *p* < 0.001). Application of 0.1 µM menthol in all tests caused a rapid hyperpolarization of membrane potential which was accompanied with the switching off of the ability of neurons to generate action potentials (Figure 1Aa). This effect was short lasting and after some seconds (Av: 19.27 ± 4.5 s) the membrane potential returned to its control value. There was relatively large variation in the duration and size of hyperpolarization induced by menthol between preparations. To express quantitatively the changes in DUM neurons activity induced by tested substances, we decided to estimate the Relative Size of Hyperpolarization (RSH) value—this is the value of hyperpolarization (counted as each deviation from basal (control) membrane resting potential) multiplied by the duration of such hyperpolarization. In that manner, we obtained a “surface” of neuronal response for applied substances (Figure 1Ba). In the case of non-occurrence of hyperpolarization, we considered 15 s of activity after the application of tested substances or physiological saline. After application of menthol, RSH was equal to −793.39 ± 243.35 mV*s (Figure 1Bb) compare to −35.65 ± 30.55 mV*s in the control. The difference was statistically significant with *p* = 0.0037. The negative value of RSP for the control was due to the significant after-hyperpolarization (one of the phases) of each action potential.

Very similar observations were made for 0.1 µM octopamine (Figure 1(Ac,Bb)), which caused hyperpolarization related to the lack of ability to generate action potentials. RSH value after its application was equal to −705.45 ± 190.47 mV*s and the difference with the control was significant with *p* = 0.0032. There was no difference with menthol trial.

We assumed that menthol acted through the octopamine receptors. To verify that hypothesis, we performed experiments by preincubating for 5 min ganglia with phentolamine which is an octopamine receptor antagonist. Phentolamine (10 µM) completely abolished the effect of menthol as well as octopamine (Figure 1(Ab,Ad,Bb)). RSH value for menthol in the presence of phentolamine was equal to −66.55 ± 12.30 mV*s and was statistically not different from the control.

One of the possible effects of activating octopamine receptor is the triggering off the PKA signaling pathway. In the next step, we performed an electrophysiological experiment by pretreating ganglia with H-89 (1 µM) which is an inhibitor of PKA. H-89 applied 5 min before menthol, completely blocked its effect and gave an RSH value of −34.46 ± 4.43 mV*s (Figure 1Bb).

### 2.2. Calcium Imaging

The activation of octopamine receptors can lead to an increase in intracellular calcium level, therefore we performed calcium imaging on dissociated DUM neurons, as seen in Figure 2. In control conditions, the signal resulting from binding free calcium ions with Oregon BAPTA-1 indicator was very low—it was equal to 6.65 ± 0.69 MGV and was observed mainly in the vicinity of the cell membrane (Figure 2A,C). Applied substances changed calcium signal (F_2,42_ = 8.036, *p* = 0.001). Application of 0.1 µM menthol in bath resulted in the increase in calcium signal to 13.92 ± 1.95 MGV, *p* = 0.00013. The highest signal intensity was observed near the cell membrane (Figure 2B,C). Since H-89 completely blocked the effect of menthol in electrophysiological studies, we decided to test its effect on menthol-induced increase of calcium level. After pre-treating cells with the PKA inhibitor, the intensity of fluorescence was equal to 6.93 ± 0.72 MGV and did not differ from control.

### 2.3. Acetylcholinesterase Activity

In our previous work [18], in electrophysiological experiments on the cockroach escape system, we showed that menthol increased the efficacy of bendiocarb insecticide which is an AChE inhibitor. In this study, we analyzed bendiocarb efficiency in the presence of menthol on AChE enzymatic level. In control conditions, the activity of AChE obtained from dissociated terminal abdominal ganglia (TAG) of *P. americana* was equal to 1.86 ± 0.16 µmol/mg protein (Figure 3B). Menthol applied in concentration ranges 0.1 nM–100 µM did not significantly change AChE activity (Figure 3B) (F_8,16_ = 1.26, *p* = 0.33).

Tested substances changed activity of AChE enzyme (F_25,50_ = 39.19, *p* < 0.001). Bendiocarb inhibited AChE activity with ED_50_ = 0.10 µM and with a maximum effect of 88.76% of inhibition for 1 µM concentration. The lowest tested concentration of bendiocarb which caused enzyme inhibition (8.4%) was 0.05 µM (Figure 3A). Application of menthol (0.1 µM) shifted the dose-inhibition curve of bendiocarb toward lower concentrations and decreased by 36% ED_50_ value, which was equal to 0.074 µM. The maximum observed effect for 1 µM bendiocarb in the presence of menthol was 92.96% of inhibition and for the lowest tested bendiocarb concentration – 0.05 µM it was 35.7%. The increase in bendiocarb efficacy in the presence of menthol was statistically significant; activity of AChE in the presence of bendiocarb (0.5 µM) was equal to 18.87 ± 2.76% of native value and with addition of menthol (0.1 µM), it was equal to only 11.43 ± 2.62%; *p* = 0.022 (Figure 3(Aa,Ab),C).

The pre-incubation of ganglia with phentolamine (10 µM) and H-89 (1 µM) abolished the potentiating effect of menthol, leading to ED_50_ values of 0.11 and 0.12 µM for bendiocarb in the presence of menthol with the two respective inhibitors. The AChE activity in the presence of phentolamine (10 µM), menthol (0.1 µM) and bendiocarb (0.5 µM) was equal to 23.98 ± 0,99% of initial value and was higher than that of bendiocarb alone (18.87%) although not significantly. The AChE activity in the presence of H-89 (1 µM), menthol (0.1 µM), and bendiocarb (0.5 µM) was equal to 27.25 ± 2.47% of initial value and was higher than that of bendiocarb alone (18.87%) with *p* = 0.01 (Figure 3(Aa,Ab),C).

The pre-incubation of ganglia with bisindolylmaleimide IX (iPKC), an inhibitor of protein kinase C signaling pathway, did not have any effect on the action of menthol with the ED_50_ value being equal to 0.08 µM. The AChE activity in the presence of iPKC (1 µM), menthol (0.1 µM), and bendiocarb (0.5 µM) was equal to 12.89 ± 2.29% of initial value and was lower than that of bendiocarb alone (18.87%) with *p* = 0.024 (Figure 3Ab,C). The above values were similar to those of the menthol trial.

## 3. Discussion

Cooperativity between essential oils and chemical insecticides in the control of pest insects becomes more and more interesting in modern pest management. Recent studies evaluated synergistic interaction between essential oils and chemical insecticides [37,38,39].

In our previous paper [18], it was demonstrated that menthol potentiates the effect of the carbamate insecticide, bendiocarb. As a possible factor increasing the efficiency of bendiocarb, the activation of octopamine receptors by menthol in the nervous system of the cockroach was proposed. However, the mechanism of this enhancement remained unclear. The aim of our new study was to obtain a deeper insight into this mechanism. Experiments were performed on cockroach DUM neurons, which are known to be octopaminergic.

Octopamine serves as a neurotransmitter, neurohormone, and neuromodulator in insects [40,41,42,43] although it is only a “trace amine” in vertebrates [28,43,44]. Octopamine receptors are G protein-coupled receptors. Their activation turns on cellular signal transduction pathways and changes various insect organism functions. Due to their key role being limited to invertebrates, they were tested as targets for insecticides: formamidine pesticides and plant essential oils [23,45].

Sensitivity of DUM neurons to octopamine was described previously. It has been observed that the electrical activity of isolated DUM neurons is regulated by octopamine in concentration dependent manner [46,47]. In our experiments on DUM neurons in situ, the ejection of octopamine induced rapid and short-lasting hyperpolarization of membrane potential and inhibition of spontaneous discharges. After several seconds, spontaneous action potentials were again generated. Highly similar effects were observed after the application of menthol and it was abolished by phentolamine, an octopamine receptor inhibitor. Identity in effects of octopamine and menthol on DUM neurons electrical activity confirmed the statement presented in our previous paper [18] that menthol activates octopaminergic receptors.

“Pacemaker potential”—depolarizing changes of membrane potential occurring during intervals between spikes—is “driven” by several calcium ion currents: low voltage activated (LVA) Ca^2+^ currents [48]; mid/low voltage (M-LVA) activated Ca^2+^ current [47]; and high voltage activated (HVA) Ca^2+^ current [36,47,48]. Between LVA Ca^2+^ currents, the maintained current permeable to Na^+^ and Ca^2+^ (mLVA Na^+^/Ca^2+^) has been described in DUM neurons [49]. It plays an essential role in regulating DUM neurons spontaneous discharge frequency [50]. Study of Lapied et al. [50] demonstrated that the application of octopamine on isolated DUM neurons decreased the amplitude of mLVA Na^+^/Ca^2+^ via an increase in internal cAMP level and activation of protein kinase A (PKA). PKA activation induces a negative regulatory action on mLVA Na^+^/Ca^2+^ by phosphorylation of DARPP-32 protein. The effect of octopamine on mLVA Na^+^/Ca^2+^ was abolished by phentolamine.

We propose the mLVA Na^+^/Ca^2+^ current to be one of the targets for the menthol action. The PKA inhibitor (H-89) has been shown to reverse the effects of octopamine on mLVA Na^+^/Ca^2+^ current [50]. In our experiments, applying of H-89 in the bath eliminated the effect of menthol on the electrical activity of DUM neurons. The stimulation of octopamine receptors in *Periplaneta americana* causes the activation of adenylyl cyclase, increase of cAMP level, and activation of PKA [29]—factors which reduce the amplitude of mLVA Na^+^/Ca^2+^ current [50]. When the depolarizing current was inhibited, hyperpolarization should occur—this was observed in our experiments. After blocking the PKA, its negative regulatory action was abolished and the effect of menthol was no longer observed.

The hyperpolarization state induced by octopamine and menthol can lead to activation of tLVA Ca^2+^, which is normally activated at membrane potential of -70 mV and is involved in the initial part of the pre-depolarization phase of the activity of DUM neurons [48]. Depolarizing tLVA Ca^2+^ may cause a return of membrane potential from deep hyperpolarization to normal potential level after octopamine and menthol. Moreover, the hyperpolarization induced by octopamine and menthol can also increase the resting Ca^2+^ current that will participate in the return to the inter-spikes membrane potential level [51].

Activation of octopamine receptors, (Pa oa_1_) from *Periplaneta americana*, OAMB from *Drosophila melanogaster*, and CsOA1 from hemocytes of *Chilo suppressalis*—induces an increase in both cAMP level and internal Ca^2+^ concentration [29,52,53,54]. In our experiments, we measured the intracellular level of Ca^2+^ in DUM neurons after applying menthol and observed a significant increase in its level. Intracellular Ca^2+^ concentration was indicated as a „key factor” in the regulation of ability to generate spontaneous action potentials in DUM neuron [36]. Elevated Ca^2+^ concentration after the activation of octopamine receptors by menthol could induce modulatory effects on DUM neuron membrane ionic channels.

We consider the rise in Ca^2+^ level as secondary effect, not resulting from the direct activity of octopamine receptors. The increase in calcium level as a result of octopamine receptor has been evidenced [29,55,56], although we observed calcium rise which was blocked by protein kinase A inhibitor. We assume that menthol causes the activation of adenylyl cyclase and thus increase in cAMP level which activates PKA. The activities of PKA and adenylyl cyclase are highly regulated; cAMP can directly open the non-selective cation channels and cause Ca^2+^ to enter the cells [57,58]. The role of increasing level of Ca^2+^ ions is the regulation of adenylyl cyclase activity and thus the constitution of a negative feedback for PKA [59,60,61,62]. PKA is also responsible for up-regulation of M-LVA channel and voltage-independent Ca^2+^ resting current and thus increase the amount of calcium entering the cells which can be a reason for the rise in PKA-dependent calcium [63].

On the other hand, increase in Ca^2+^ level could indicate the involvement of PKC in the action of menthol. However, there is a lot of data indicating that PKA and PKC pathways exclude themselves and negatively regulates each other [64,65,66,67] and that cAMP inhibits PKC [68,69]. Our biochemical experiments (discussed later) confirm that PKC is not involved in the described mechanism. The inhibitor of PKC did not modify the effect of menthol on efficiency of bendiocarb.

In toxicity tests on the cockroach and experiments on its whole nervous system done during our previous study, we demonstrated that menthol increases the activity of bendiocarb—a carbamate—as an insecticide [18]. The experiments carried out on DUM neurons presented in this paper, allowed us to conclude that menthol activates the PKA signaling pathway through octopamine receptors and also helped demonstrate an increase in Ca^2+^ concentration. The modifications of the sensitivity of various molecular targets to insecticides by phosphorylation/dephosphorylation processes are already known [13,16,17,70,71,72]. The aim of the last part of our study was to determine the modification of bendiocarb efficiency by menthol at the level of enzymatic activity of AChE. In biochemical experiments, presence of menthol decreased ED_50_ of bendiocarb by 36% which closely correlates with results obtained in cockroach toxicity tests and in tests on its whole nerve cord [18]. Phentolamine eliminated the menthol-induced potentiation, which clearly indicates the involvement of octopamine receptors in its action. Moreover, bendiocarb was less potent in the presence of octopamine receptor antagonist (however, not significantly), which can indicate that some small level of octopamine is essential for normal activity of the enzyme. It is consistent with our previous results where phentolamine completely blocked the effect of bendiocarb on the whole nerve cord preparation [18]. The use of PKC inhibitor did not modify the effect of bendiocarb applied together with menthol. This suggests that the PLC signaling pathway is not involved in the straightening effect of menthol. However, the PKA inhibitor completely eliminated the effect of menthol and significantly reduced the efficiency of bendiocarb applied alone. These results clearly indicate that the potentiating effect of menthol is accomplished by the PKA signaling pathway and that phosphorylation of AChE is necessary to achieve the enhancement of bendiocarb effect by menthol. Some reports indicate that PKA, but not PKC, is responsible for phosphorylation of AChE [73].

The decrease in inhibitory activity of bendiocarb on AChE in the presence of H-89 compared to the action of bendiocarb alone confirms the conclusion from experiments with phentolamine. It can be explained by the fact, that some state of phosphorylation of AChE is required for its sensitivity for modulators, such as bendiocarb.

In this paper, we have confirmed the hypothesis that menthol increases bendiocarb efficacy and that its action is mediated through an octopamine receptor. We have shown first evidences that protein kinase A is involved in the menthol effect. We have also shown that menthol increases Ca^2+^ level and finally that protein kinase C does not participate in this action of menthol. Obtained results can be used in development of modern insecticide formulas, less harmful for human and non-target animals and more potent against insect-pests.

## 4. Materials and Methods

### 4.1. Insects

All experiments were performed on preparations from males of the cockroach *Periplaneta americana*. The insects were obtained from our breeding program. The animals were kept in temperature of 27–29 °C with access to water and food (cat food, oatmeal and apples) ad libitum. Insects were transferred to the laboratory 24 h before experiment for adaptation.

### 4.2. Reagents

Bendiocarb (Pestanal, analytical standard), dl-octopamine hydrochloride (≥95.0%), phentolamine hydrochloride (≥98.0%, TLC), and (±)-menthol (racemic ≥ 98.0%) were purchased from Sigma Aldrich (Saint Luis, MI, USA). They were dissolved in ethanol to a concentration of 10 mM and next serial dilutions were made in physiological saline. H-89 and bisindolylmaleimide IX were purchased from Abcam (Cambridge, United Kingdom). Chemicals were dissolved in DMSO to concentration 10 mM and then the serial dilutions were prepared in physiological saline.

Physiological saline contained in mM: NaCl – 210, KCl – 3.1, CaCl_2_ – 5, MgCl_2_ – 5.4, and Hepes – 5. The pH = 7.2 was adjusted with NaOH. Hepes were purchased from Sigma Aldrich. The physiological saline components—NaCl, KCl, MgCl_2_, CaCl_2_, ethanol 96%, and DMSO—were obtained from Polskie Odczynniki Chemiczne SA (Gliwice, Poland).

Acetyltiocholine chloride (99%, TLC, Sigma Aldrich) was dissolved in physiological saline to 0.1 mM. The solution stopping AChE reaction was composed of 1 mM 5,5-dithio-bis-(2-nitrobenzoic acid) (DTNB) (Thermo Fisher, Waltham, MA, USA) and 2% sodium dodecyl sulfate (SDS) (Thermo Fisher) diluted in water. Collagenase (from *Clostridium histolyticum*, type XI) was purchased from Sigma Aldrich and was diluted to final concentration 2 mg/mL in physiological saline. Streptomycin/penicillin (Roche) (50,000 U/mL penicillin G and 50 mg/mL streptomycin (as sulfate) in 0.9% NaCl) were dissolved in physiological saline to working solutions of 100 IU/mL penicillin, 100 μg/mL streptomycin. Bovine serum (Sigma Aldrich) was diluted to 1% in physiological saline. Oregon Green BAPTA-1 (Thermo Fisher) was diluted in DMSO to 40 µM stock solution and then diluted in physiological saline to a working solution of 5 µM.

### 4.3. Electrophysiological Experiments

Electrophysiological recordings were performed on DUM neurons in situ in terminal abdominal ganglia (TAG) from the ventral nerve cord of the insect. The ganglia were dissected with micro-scissors. The protein-lipid sheets covering the ganglia were removed to allow chemicals to get to the neurons inside the ganglia and facilitate insertion of the microelectrodes. The TAG were then moved to Petri dishes filled with sylgard-polymer, to which the TAG were mounted with entomological needles. The ganglia were covered with physiological saline all the time. Recordings of bioelectric signals in DUM neurons were performed using glass microelectrodes with resistance equal to 30–40 MΩ. The electrophysiological set-up was consisted of: registration microelectrode filled with 3 M KCl, reference electrode placed in physiological saline near the preparation, headstage (HS-2A, Gain: 10 MGU; Axon Instrument’s, Sunnyvale, CA, USA), amplifier Axoclamp 2A (Axon Instrument’s) and oscilloscope (Hameg HM 507). Recordings were transferred to a computer and processed by modified Hameg software (version 6.0, Toruń, Poland).

In each experiment, the control recordings were made (for 5 min) after 10 min of preparation stabilization when spontaneously generated discharges had been stable in amplitude and frequency and the membrane potential had changed only in a range of 2–3 mV. Then tested substances (menthol and octopamine) were applied by fast ejection (Picoliter Microinjector PLI-100A, Warner Instruments, Holliston, MA, USA) in close vicinity of the ganglia to obtain the final 0.1 µM concentration of substances. In the case of preincubation, phentolamine (10 µM) H-89 (1 µM) were slowly applied by perfusion 5 min before introducing the tested substances.

The collected data was further analyzed using R software [74]. The results were expressed as mean values ± SE and the comparison of several data groups was made using one-way ANOVA. The differences between groups were tested by Tukey’s post-hoc tests.

### 4.4. Acetylcholinesterase Activity—Biochemical Tests

AChE activity was determined using modified Ellman’s method [75]. For biochemical experiments, 6 TAGs were dissected and placed on ice in tube containing 2 mL of physiological saline. The ganglia were then incubated with Collagenase IX in concentration of 2 mg/mL for 30 min in temperature of 30 °C. After incubation, the ganglia were transferred to cold (4 °C) physiological saline and rinsed twice to stop collagenase enzymatic digestion. Depending on the combination of tested substances, the different samples (shown in Table 1) of ganglia were prepared. In the next step, dissociation of already treated ganglia were made – they were passed through series of glass pasteur pipets with decreasing diameters since tissue fragments were no longer visible. Solution prepared in this way contained some whole nerve cells and fragments of cells with AChE enzyme activity. 90 µL of solution was placed in each well in 96-well plates with flat bottoms on ice. Solution containing AChE enzyme was incubated with the substrate, acetyltiocholine for 30 min in temperature of 30 °C. The reaction was stopped by 1 mM 5,5-dithio-bis-(2-nitrobenzoic acid) (DTNB) with 2% sodium dodecyl sulfate (SDS). In variants of the experiments where efficiency of bendiocarb was estimated, tissue solution was incubated with insecticide 15 min before starting the reaction with acetyltiocholine. The amount of protein in solution was determined using Bradford method [76]. Calibration curve was always prepared at the same time as experiment. 10 min after stopping the reaction, the 96-well plate was read in microplate reader (BioTek, Epoch, Winooski, VT, USA) with wave lengths of 406 nm for Ellman’s reaction and 595 for Bradford’s reaction.

Each replication was prepared using 6 ganglia. The experiment was replicated 3 times. The collected data was further analyzed using R software [74]. The results were expressed as mean values (from 3 replications) ± SE and the comparison of several data groups was made using one-way ANOVA. The differences between groups were tested by Tukey’s post-hoc tests. The dose-response curves were established using ‘drc’, ‘sandwich’ and ‘lmtest’ [77,78,79].

### 4.5. Calcium Imaging

The determination of calcium level in living cells was performed on dissociated DUM neurons. 6 ganglia were dissected and placed in a tube containing 2 mL of physiological saline with streptomycin/penicillin in sterile conditions. The ganglia were then incubated with sterile Collagenase IX in concentration of 2 mg/mL for 30 min in temperature of 30 °C. The enzymatic activity was stopped by rinsing the ganglia with sterile physiological saline containing antibiotics. The ganglia were then moved to the physiological saline containing bovine serum and antibiotics. In the next step, the ganglia were dissociated as previous described, however, in sterile conditions all the time. Obtained medium with DUM cells was divided into 2 Corning Petri dishes (covered earlier with poli-d-lysine (50 µg/mL)) and filled with sterile medium to 2 mL. The cells were incubated in temperature of 30 °C for 24 h to ensure good adhesion. After 24 h physiological saline with bovine serum and antibiotics was removed and the DUM cells were covered by sterile physiological saline.

Fluorescent marker of free calcium ions, Oregon Green BAPTA-1 at a concentration of 5 µM was applied for 1 h in the dark. Next, the liquid was removed, and cells were rinsed with fresh physiological saline. Firstly, the control observations were made. Menthol 0.1 µM was then applied in the bath and after 15 min the cells were observed. In case of pre-incubation with H-89 1 µM, the inhibitor was applied 10 min before menthol. The images were captured with a Leica TCS SP8 confocal microscope using an argon-ion laser emitting light at a wavelength of 488 nm (408 nm excitation wave). Optimized pinhole, long exposure time (400 kHz), and 20X (numerical aperture, 1.4) Plan Apochromat DIC H lens were used. For the quantitative measurements, each experiment was performed under condition of consistent temperature, incubation time, and concentrations of probes. The images were collected under consistent conditions of acquisition (low laser power at 3%, emission band, gain, and resolution) to ensure comparable results. For bleed-through analysis and control experiments, LASX program (Leica Application Suite X) software was used. Between 10 and 20 cells were analyzed for each experimental variant. The level of fluorescence was expressed in arbitrary units (as the mean intensity per μm^2^). The statistical analysis was performed using SigmaPlot 11.0 software. The statistical significance was determined by one-way ANOVA followed by Tukey’s post-hoc test (*p* < 0.01 or *p* < 0.001) to compare effects of different treatments.

## Figures and Tables

**Figure 1 molecules-24-03775-f001:**
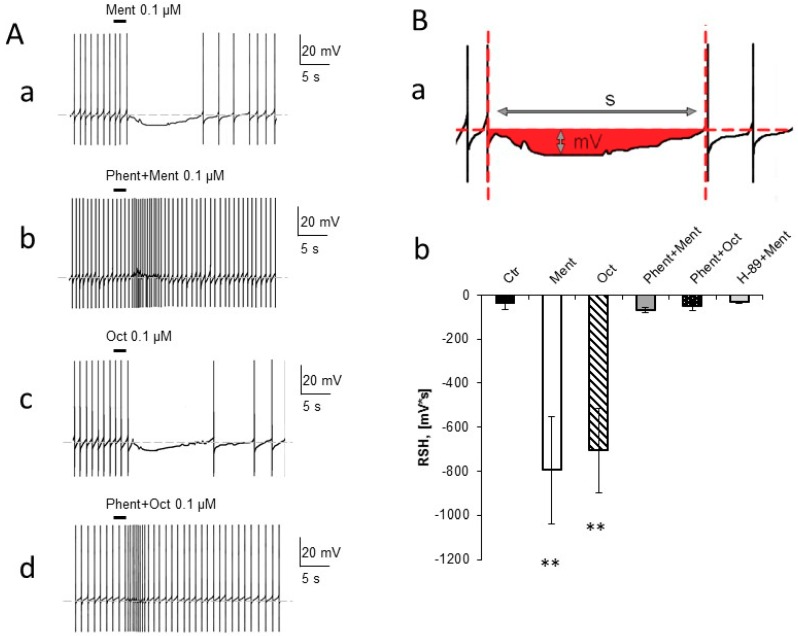
Menthol and octopamine change the electrophysiological properties of dorsal unpaired median (DUM) neurons. (**A**) Representative original recordings of spontaneous action potentials of DUM neurons: (**a**) deep hyperpolarization and switching off of the spontaneous potentials as a result of the application of menthol (0.1 µM); (**b**) phentolamine (10 µM) blocked the hyperpolarization caused by menthol; (**c**) octopamine (0.1 µM) caused the same effect as menthol; (**d**) phentolamine (10 µM) blocked the hyperpolarization caused by octopamine. (**B**) (**a**) The representation of RSH (Relative Size of Hyperpolarization) value. Size of DUM neurons response (shown in red) to menthol and octopamine application was expressed quantitatively by RSH—value of hyperpolarization (mV) was multiplied by time (s) of its duration to obtain a response surface area. (**b**) Effect of menthol (0.1 µM, Ment) and octopamine (0.1 µM, Oct) which were reversed by phentolamine (10 µM, Phent + Ment; Phent + Oct). Inhibitor of PKA – H-89 (1 µM) abolished the hyperpolarization caused by menthol (H-89 + Menth). High negative values correspond to deep and long hyperpolarization. The data is presented as mean values ± SE, *n* = 10. The statistically significant differences between control and tested substances are marked: ** *p* < 0.01.

**Figure 2 molecules-24-03775-f002:**
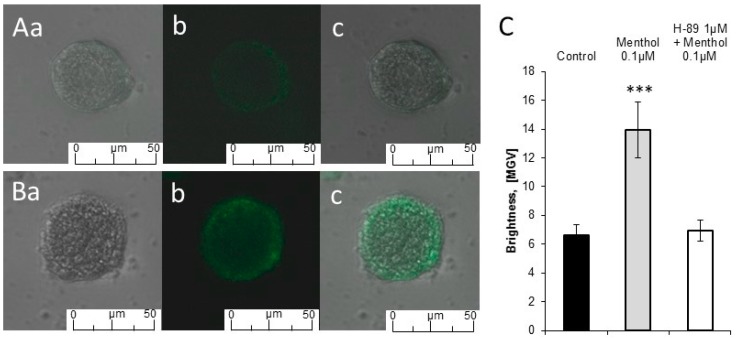
Menthol changed the level of free calcium ions in DUM cells. The free calcium ions were labeled by Oregon Green BAPTA-1 and observation of fluorescence of the living cells was made using a confocal microscope Leica TCS SP8 (excitation at 308 nm, detection at 488 nm - green fluorescence). (**A**) Control confocal fluorescence images of DUM neuron and (**B**) after incubation with menthol 0.1 µM. (**a**) The bright-field transmission images; (**b**) the fluorescence transmission images; and (**c**) merged transmission images of both control and treated DUM neurons respectively. Note a high discrepancy between signal in control neurons and after application of menthol. Scale bars are 50 μm. (**C**) Numerical representation of free calcium level in DUM neurons in control, after application of menthol 0.1 µM alone, and preincubated with H-89. The data is presented as mean values ± SE. The statistically significant differences between control and tested substances are marked: *** *p* < 0.001.

**Figure 3 molecules-24-03775-f003:**
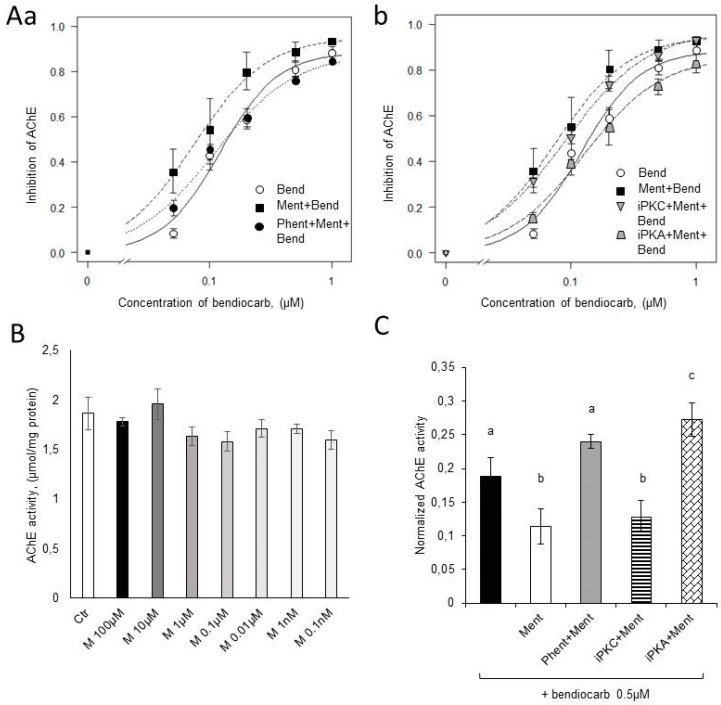
Biochemical analysis of acetylcholinesteraze (AChE) activity and its inhibition by bendiocarb insecticide. (**A**) Dose-inhibition curves representing inhibitory effect of bendiocarb on AChE activity: (**a**) Dose-dependency of AChE inhibition caused by bendiocarb (Bend) was shifted to its lower concentrations in the presence of menthol (0.1 µM, Ment + Bend). The effect of menthol was reversed by phentolamine (10 mM, Phent + Ment + Bend); (**b**) Dose-dependency of AChE inhibition caused by bendiocarb (Bend) was shifted to its lower concentrations in the presence of menthol (0.1 µM, Ment + Bend). The effect was reversed by protein kinase A inhibitor, H-89 (1 µM, iPKA + Ment + Bend) but not by protein kinase C inhibitor, bisindolylmaleimide IX (1 µM, iPKC + Ment + Bend). (**B**) Menthol applied in different concentrations (M) did not changed the AChE activity compared to control (Ctr). (**C**) Inhibition of AChE activity by bendiocarb (0.5 µM, black bar) and the changes caused by: menthol (0.1 µM, Ment); phentolamine (10 µM) with menthol (0.1 µM, Phent + Ment); inhibitor of protein kinase C, bisindolylmaleimide IX (1 µM) with menthol (0.1 µM, iPKC + Ment); and the inhibitor of protein kinase A, H-89 (1 µM) with menthol (0.1 µM, iPKA + Ment). The different letters above the bars refer to statistically significant differences between the data with *p* < 0.05.

**Table 1 molecules-24-03775-t001:** Samples composition in biochemical tests.

Sample Name	Physiological Saline	Menthol (µM)	Bendiocarb (µM)	Phentolamine (µM)	iPKC (µM)	iPKA (µM)
Control	+					
Ment		0.0001, 0.001, 0.01, 0.1, 1, 10, 100				
Bend			0.05, 0.1, 0.2, 0.5, 1			
Ment + Bend		0.1	0.05, 0.1, 0.2, 0.5, 1			
Phent + Ment + Bend		0.1	0.05, 0.1, 0.2, 0.5, 1	10		
iPKC + Ment + Bend		0.1	0.05, 0.1, 0.2, 0.5, 1		1	
iPKA + Ment + Bend		0.1	0.05, 0.1, 0.2, 0.5, 1			1

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
