# Peer review of "Menthol Increases Bendiocarb Efficacy Through Activation of Octopamine Receptors and Protein Kinase A"

_molecules, 2019, doi:10.3390/molecules24203775_

Round 1

Reviewer 1 Report

Why did author test synergistic effect of menthol on carbamates?No synergistic effect on pyrethroids?  Add F value and degree of freedom for statistical analysis. In Figure 3 A a and b : Standard error bar is necessary 

Reviewer 2 Report

It is very important to find a effective strategy for plant protection. This manuscript is a continuation of the author's previous publication. And it is of some significance to study the mechanism that the menthol increases bendiocarb efficacy. However, in this manuscript, menthol had no inhibitory effect on AChE at the concentration range of 0.1-100 μM. Based on the published paper (Experimental and Applied Acarology 2010, 52 (3) , 261-274. DOI: 10.1007/s10493-0), the inhibitory rate of menthol on AChE at the concentration of 10 μM was 66.6%. And their test method refers to the same method. Therefore, it is not clear whether the inhibition of AChE by menthol and bendiocarb is synergistic or additive.  

Reviewer 3 Report

The aim of this study was to verify the hypothesis that menthol potentiates the effect bendiocarb through the activation of octopamine receptors and thus the activation of phosphorylation cascade. Octopaminergic dorsal unpaired median (DUM) neurons of the cockroach (Periplaneta americana) nervous system was taken as a model pest. It is a relatively well prepared MS that is suitable for publishing in Molecules, following minor modifications.

1. In the introduction, better justify the choice of chemicals (a.i. of insecticides), especially in terms of environmental and health safety. It is known that Bendiocarb is highly toxic to birds and fish. In general, Carbamate insecticides target human melatonin receptors, along with inhibiting acetylcholinesterase. It is therefore important at the outset to better justify the choice of these substances in terms of the possible implementation of research results into practice.

2. In recent years, several studies have dealt with the synergistic and antagonistic relationships of substances contained in essential oils against insects. This phenomenon should be mentioned in the discussion.

3. In conclusion discuss the possible implementation of the results achieved in to practice.
